# People's naïve belief about curiosity and interest: A qualitative study

**Sumeyye Aslan**[ID]*, **Greta Fastrich**[¤a], **Ed Donnellan**, **Daniel J. W. Jones**, **Kou Murayama**[¤b]*

School of Psychology and Clinical Language Sciences, University of Reading, Reading, England, United Kingdom

¤a Current address: School of Psychological Science, University of Western Australia, Perth, Australia
¤b Current address: Hector Research Institute of Education Sciences and Psychology, University of Tübingen, Tübingen, Germany
* sumeyye.aslan@pgr.reading.ac.uk (SA); k.murayama@reading.ac.uk (KM)

## Abstract

The purpose of this study was to critically examine how people perceive the definitions, differences and similarities of interest and curiosity, and address the subjective boundaries between interest and curiosity. We used a qualitative research approach given the research questions and the goal to develop an in-depth understanding of people's meaning of interest and curiosity. We used data from a sample of 126 U.S. adults (48.5% male) recruited through Amazon's Mechanical Turk ($M_{age}$ = 40.7, $SD_{age}$ = 11.7). Semi-structured questions were used and thematic analysis was applied. The results showed two themes relating to differences between curiosity and interest; active/stable feelings and certainty/uncertainty. Curiosity was defined as an active feeling (more specifically a first, fleeting feeling) and a child-like emotion that often involves a strong urge to think actively and differently, whereas interest was described as stable and sustainable feeling, which is characterized as involved engagement and personal preferences (e.g., hobbies). In addition, participants related curiosity to uncertainty, e.g., trying new things and risk-taking behaviour. Certainty, on the other hand, was deemed as an important component in the definition of interest, which helps individuals acquire deep knowledge. Both curiosity and interest were reported to be innate and positive feelings that support motivation and knowledge-seeking during the learning process.

## Introduction

The concepts of curiosity and interest have both received increasing attention in the literature of motivation and education [1]. Curiosity has been shown to positively affect learning outcomes and processes [2]. Curiosity is related to higher academic performance on standardized tests [3] and academic persistence [4]. Likewise, interest supports learning in different content areas [5], and has been shown to facilitate a critical cognitive and affective motivational variable that aids attention [6]. Despite an increasing amount of work on these topics in recent years, a critical issue remains in the field: lack of consensus about how researchers should conceptualize these terms [7–12]. Although there is a view that we do not necessarily need to

**Data Availability Statement:** All relevant data are within the manuscript and its Supporting Information files.

**Funding:** This research was in part supported by JSPS KAKENHI (Grant Numbers 18H01102;

18K18696), the Leverhulme Trust (Grant Number RL-2016-030), Jacobs Foundation Advanced Fellowship, and the Alexander von Humboldt Foundation (the Alexander von Humboldt Professorship endowed by the German Federal Ministry of Education and Research). KM received all of the awards. The funders had no role in study design, data collection and analysis, decision to publish, or preparation of the manuscript.

**Competing interests:** The authors have declared that no competing interests exist.

strictly define these terms to progress the field [13, 14], inconsistent use of the language may hinder effective scientific communications among researchers [8–15]. In addition, thinking about the definitions of these terms may shed new light on the psychological processes underlying the states of curiosity and interest.

In the classical literature, curiosity has been explained within the framework of drive reduction theories, treating it like an appetite analogous to other primary needs such as hunger, [16]. Further it is proposed that curiosity is the pleasant experience of novelty seeking and as such is an optimal arousal state lying in between feelings of anxiety and boredom [17]. Additionally, Loewenstein describes curiosity as associated with the identification of unknown pieces of information (i.e. knowledge gap) [7]. Another approach states that a dynamic subsystem regulates attentional focus through a spontaneous learning process, thus curiosity is the part of a larger unconscious mechanism [18].

Research on interest has been mainly developed in the literature of educational psychology. Similar to research on curiosity, there have been different perspectives on how interest should be conceptualized and theorized. For example, the four phase model of interest development, one of the most prominent interest theories in the literature, supposes that there are four phases in the development of interest: (1) triggered situational interest; (2) maintained situational interest; (3) emerging individual interest; and (4) well-developed individual interest [19]. The distinction between situational interest (1 & 2) and individual interest (3 & 4) is critical to the model. Situational interest is conceptualized as focused attention triggered by environmental stimuli, individual interest is conceptualized as a predisposition for reengaging with a particular topic. Krapp conceptualizes interest by connecting it to the person's growing awareness of the self. A critical element in his conceptualization is the integration of ones' self and the activities one is interested in [20]. The Expectancy-value Theory [21, 22] examines task interest by considering the influence of the subjective value of certain activities on motivation and achievement in school. Interest is generally examined by considering individual differences in engagement with educational activities and motivation. Another prominent model of interest, the self-regulation of motivation model [23], conceptualizes interest as a resource for self-regulation and focuses on people's ability to self-generate interest.

Distinguishing curiosity and interest is the subject of much discussion. One common perspective is that curiosity is a momentary motivation to explore novel or puzzling phenomena [24, 25], whereas interest represents a more long-term developmental process, with emphasis on its stability in personality [26]. In other words, curiosity is sometimes conceptualized as immediate experiences in response to stimuli in the external environment (e.g., novel puzzles) while interest represents more active engagement within a learning context [19]. However, it is still unclear how in-the-moment experiences generated from interest (often called situational interest) are related to curiosity. In addition, this idea does not clearly explain how trait-level curiosity (i.e., trait curiosity; [27]) is distinguished from developed interest. As noted by Ainley, shared underlying characteristics of curiosity and interest (e.g., attentional processes and exploratory behaviour) are intertangled in infancy and early childhood; however, the experiential states associated with trait curiosity and interest diverge in later educational contexts [10].

Another discussion about curiosity and interest concerns whether they are instinctive (externally instigated by one's environment) or intentional (internally instigated; [28]). While children's curiosity is characterised as an exploratory behaviour, especially between ages 4 and 6 years, individual interests (dispositions towards reengaging with specific topics) do not arise until adolescence or adulthood [29]. From childhood to adulthood therefore, curiosity and interest have been reflected differently: in childhood curiosity is viewed as instinctive exploratory behaviour in response to external stimuli, and in adulthood interest is seen as intentional, self-motivated exploration. However, Peterson and Cohen define domain-specific curiosity as

an active and intentional experience [30] and curiosity is defined as a conscious investigation and unspecified exploration of the environment [14]. Therefore, the terms can both be used to refer to intentional, internally-motivated behaviour or instinctive, externally-motivated behaviour.

The lack of consensus poses a challenge to empirical investigations of curiosity and interest, which need to "measure" these constructs in some way. For example, curiosity and interest are often examined together by self-reported measures. One such instance is the Interest-Deprivation Type Epistemic Curiosity scale, which distinguishes interest-type vs deprivation-type of curiosity [31]. As the name indicates, in this scale, interest is treated as one component of trait curiosity. Likewise, Litman and Spielberger, showed that interest emerges in measures of curiosity (i.e., noted by Berlyne in his definitions of epistemic curiosity, defined as a "drive to know" in the presence of a knowledge gap, and perceptual curiosity, defined as "the curiosity which leads perception of stimuli" via the tactile stimulation of humans and animals) such that curiosity is similar to triggered interest [32]. However, it is argued that interest has a broader range of triggering variables not limited to collative ones [1–8]. Interest is also often assessed by a subscale of intrinsic motivation [33] but the boundary between intrinsic motivation and interest is also unclear. Despite numerous self-report measures (e.g., State-Trait Personality Inventory; [34], Curiosity/interest in the World; [35], Epistemic Curiosity Scale; [27]) as well as task-based assessments/manipulations (e.g., trivia questions), there are no agreed-upon measures of curiosity and interest: they are generally based on subjective judgements or interpretations of researchers [28]. The issue is not limited to self-reported measures. Even for studies that use physiological or neuroscientific methods (e.g., eye-tracking, functional magnetic resonance imaging; [36, 37]), there are no agreed-upon indicators of curiosity and/or interest.

The long-lasting debate, disagreement, and confusion about the definition of curiosity and interest poses a fundamental question: *Why* is it so difficult to define curiosity and/or interest? A reward-learning framework of knowledge acquisition [14], which was recently proposed to explain the mechanistic process underlying knowledge acquisition behavior, could give us a clue. According to the reward-learning framework, people seek information as a consequence of the reward-learning (reinforcement-learning) process—the framework presupposes that information is inherently rewarding and, as a result, people expect information-seeking behavior to bring about reward. This inherent reward value of information may be driven by uncertainty (or because acquiring it resolves a knowledge gap). In addition, the framework clarifies the mechanisms by which people establish a long-term commitment to information seeking, explaining why people engage with a certain topic for a long period of time without any extrinsic rewards. Specifically, the framework suggests that increased knowledge enhances both the expected value of new upcoming information and people's ability to understand it, creating a positive feedback loop of information-seeking behavior. Importantly, the framework explains various behaviors that people typically attribute to "curiosity" and "interest" without using such constructs (e.g., the framework does not suppose any elements of "curiosity" or "interest" in the mental process). These behaviors are solely explained by the interaction between reward-learning and knowledge expansion processes. Given this, what are curiosity or interest? According to the reward-learning framework, curiosity and interest are "subjective psychological constructions" of these mental processes [14]. People do not typically have direct access to the reward-learning process underlying knowledge acquisition behavior. However, the underlying reward-learning process produces a myriad of subjective experiences, and from these subjective feelings, people psychologically "construct" various concepts/languages that are useful to explain their subjective feelings and behavior [38]. Curiosity and interest are considered as two such concepts/languages.

This framework effectively explains the fundamental difficulty of defining curiosity and interest: They are hard to define because curiosity and interest are overly subjective categorization of ambiguous feelings. In fact, curiosity and interest are naïve concepts that laypeople have used long before the scientific investigation of these concepts started. Like much other lay language (e.g. "enjoyment") such naïve terms do not have to have strict scientific definitions. Therefore, while we may be able to discuss and develop an agreed-upon definition, we should not expect that there are *correct* definitions of curiosity and interest. In other words, if there are two different perspectives to define these terms, there is no way to decide which is right or not. Note that this idea is not novel—in other fields (e.g., emotion research), psychological construction is a major perspective used to understand psychological constructs [39]. This perspective is also quite popular in philosophy of mind (e.g., scientific realism and the plasticity of mind, [40]).

At the same time however, because these terms are developed separately in daily language, it is likely that this results from people having somewhat distinct feelings (albeit some overlap) relating to knowledge acquisition, i.e., different experiences of their knowledge acquisition process. Our daily language does not develop in a vacuum—if lay people can distinguish two concepts, there is good possibility that these two concepts reflect (at least partly) different psychological processes [14]. Therefore, although lay people do not have direct access to precise mental processes, scrutinizing lay people's definitions of curiosity and interest may provide us with some insights into how the knowledge acquisition process is organized in our mind. Examining lay perspectives may also provide scientific researchers with a good basis to establish agreed-upon scientific conceptualizations of curiosity and interest if they wish. If scientific definitions of curiosity and interest deviate substantially from what lay people believe they are, for example, researchers (especially researchers in applied fields) would have difficulty in effectively communicating their ideas with the general public. In that case, there is no logical reason for researchers to label them as "curiosity" and "interest" (i.e., researchers should use different technical terminologies). It is also worth noting that lay people's belief is often a powerful predictor of their behavior, because people make a decision based on their own belief, not on an externally defined scientific concept [41–43]. As such, understanding people's beliefs about curiosity and interest could help us develop a theory that predicts their behavior in their daily life.

The purpose of the current study is to conduct a preliminary and exploratory investigation on how lay people define curiosity and interest. We asked participants to define curiosity and interest in their own words, and examined the similarities and differences of these terms in a bottom-up manner. There are several studies that examined people's perceptions about curiosity/interest [43, 44]. Kashdan et al. examined the relationship between self-ratings and other ratings (provided by friends and parents) of a person's curiosity traits and found moderate correlations between them, indicating that people have a common idea of what curiosity means [44]. Thoman et al. examined people's implicit theory of interest regulation—whether people believe that they can change and regulate their own interest or not [43]. They found that people who believe that interest can be changeable used more interest-regulation strategies when faced with a boring task, suggesting the importance of lay-people's beliefs about curiosity/interest. Most closely related to our work, Post and Walma van der Molen. interviewed school-children asking them to explain what curiosity means to them [45]. They found that children's conception of curiosity mainly concerns the social domain (e.g., other people's private affairs), rather than the educational domain (which the concept of interest may be more strongly related to). However, there has been a lack of research that directly examined potential commonalities and differences in lay people's perception of curiosity and interest. Because participants were not prompted in any other way than the structured open-ended question, whatever

responses the participants produced were presumed to represent their psychological reality, without using any follow-up questions. A qualitative approach may be one of the best ways to capture the definition and interpretation from these free texts. Qualitative approaches have become increasingly common in social studies, and researchers have proposed various different methodologies (e.g. pragmatism of the pluralistic approach, interpreting data pluralistically see [46]). Qualitative techniques provide rich data to evaluate studies and generate hypotheses [47] and give researchers an opportunity to understand the concepts deeply [48]. Unlike quantitative methods, the focus is not on confirming a theory—rather, the purpose is to generate hypotheses by valuing the idiosyncratic interpretations of researchers. We used thematic analysis from Braun and Clarke's approach [49] to ensure an in-depth exploration of the data, whilst enabling the research to capture a breadth and diversity of views.

## Methods

The study was approved by the University of Reading Research Ethics Committee (reference number 2016-109-KM).

### Participants

We recruited 135 U.S. adults (including nine participants recruited in our pilot study) from Amazon's Mechanical Turk (i.e., MTurk) between July and August 2018 (participants were paid $1 for study completion). The data include nine participants from a pilot study. This pilot study was conducted to ensure the quality of the data before running the study with a large number of participants. These participants answered exactly the same questions as participants in the main study. To increase the data quality on MTurk [45], we put short questions about participants' attention and whether they cheated (i.e., looked up definitions of curiosity/interest) during the experiment (we emphasized that responses to these questions would not influence payment). In addition, before analysing the data, we checked the quality of participants' responses. As a result, nine participants were excluded for copying and pasting the same responses for multiple questions or admitted to checking the internet to answer questions. Note that there were more six participants who only partially completed the experiment. After checking the authenticity of their responses, we decided to include these participants in order to make full use of the collected data. Thus, the final sample consisted of 126 participants (48.50% male). Inclusion criteria were that participants were English-speaking, had access to a computer, and lived in the United States. English was the participants' mother tongue with the exception of ten participants, who started to speak English between 1–10 years old. As for their highest educational qualifications, 40.47% had a university undergraduate degree, 18.25% had a postgraduate degree, 13.49% completed GCSEs, 13.49% completed A levels, and 3.9% indicated that they had no formal education. Ages ranged between 23 and 72 years old, mean of 40.70 years old ($SD$ = 11.70). In terms of race/ethnicity (not allowing multiple selections), 66.90% endorsed Caucasian, 4.60% endorsed African-American, 20% endorsed Asian/Pacific Islander, 4.60% endorsed Native American, 2.30% endorsed Hispanic, and 1.60% endorsed other. The data (excluding the pilot data) was also used as part of a separate, quantitative project [11] but was not analysed using qualitative methods, and did not utilise responses to all questions used in the current study.

The sample was not representative: convenience sampling was used [50, 51]. The sampling enabled different naïve beliefs regarding curiosity and interest to be uncovered [52]. Theoretical saturation, which refers to the point at which the collection of additional data adds little or nothing new for the study, is broadly accepted to reach sufficient data [53] and external validation [54] in qualitative research. We recruited 135 participants due to budgetary reasons, and

then analyzed and checked the data to see whether the data showed thematic saturation [55]; it indeed did, and therefore data collection was stopped.

## Data collection

Before starting the study, the participants read and clicked to confirm the consent form on the screen. In the consent form was indicated that their data will be used anonymously for this research. Also, the participants filled the demographic questions without their names after the consent form and their participation numbers was only used in the results. The study description was purposefully vague to prevent demand characteristics. The study description stated "In this study, you will simply answer open questions and a short questionnaire about motivation. The purpose of the study is to understand your naïve perceptions regarding motivation. Please answer these questions with your own words (please do not check internet etc.), as there is no right answer to these questions. There is no word limit. However, please answer each question with no less than 80 words."

The open-ended questions appeared on the screen after short demographic questions. There were three questions: "How do you define 'curiosity'? (i.e. being curious about X; feeling curious)"; How do you define 'interest'? (i.e. being interested in X; feeling interesting); "What do you think about the similarities and differences between 'curiosity' and 'interest'?". The order was fixed across all participants. There was no time limit. As can be seen in the instructions, we tried to minimize the potential demand characteristics bias by simply asking these questions in a neutral manner, without suggesting that these concepts are different. When responding to the first question, participants also did not know that they were going to be asked about the other concept (i.e. in the first question, participants were very unlikely to mentally compare these concepts). The data showed that participants used between 80 and 150 words in response to each question, and none of the participants indicated that curiosity and interest are exactly the same concept. This is consistent with our independent data (using a similar experimental procedure) in which only 1% of the participants indicated that curiosity and interest are completely similar on a 5-point Likert scale [11].

## Data analysis

The online transcripts were imported into qualitative analysis software (NVivo 12) for thematic analysis based on Braun and Clarke's criteria [49]. There is a lack of connection with any specific ontological or epistemological position in qualitative research; researchers critically apply a post-positivist perspective [56]. Combined with this perspective, thematic analysis allows for conceptualisation of precise phenomenon and aids qualitative researchers to conduct relatively objective analysis of data [49]. We followed Braun and Clark's six phase thematic analysis method, which is comprised of constant comparative techniques [49]. These phases are summarized in the Appendixes. In phase 1, we imported data to NVIVO software, and then read and re-read the transcripts. In phase 2, we conducted line-by-line coding. After coding, we grouped similar information by using abstract labels (*i.e. **active mind** below)*. Some labels were chosen directly from the data (*i.e. **actively thinking***). For instance, the following two participants' quotes were coded as actively thinking and trying new experiences (relating to curiosity):

> "It often is good to be curious because it keeps your mind active ***(active mind)*** and you will constantly be thinking about new things ***(new)*** and different ***(different)*** things and how they all interact." (Participant 52)

"Curiosity to me is about being interesting, *(interesting)* intrigued and actively thinking *(actively thinking)* about something, someone, or someplace." (Participant 78)

Coding was an inductive and recursive process, with established comparisons applied between and within transcripts. Primarily, we coded both explicit and implicit meanings for all data. As noted above, some of the names for labels were used explicitly in participants' responses *(i.e. **curiosity is active feeling**)* but after comparing similar codes, we also generated general meanings for the codes *(i.e. **interest is stable and long term feeling for deep information**)*. For example, when we interpreted the first quote below, it suggested that curiosity is active feeling but interest is a passive feeling. However, when considering both of the following quotes altogether, we can say that the passive feeling relating to interest actually refers to having a stable long term disposition towards in-depth learning. For that reason, we put "stable feeling" instead of "passive feeling" for interest.

"Interests can also be passive (**interest is passive feeling**), where someone is content to allow the information to come to them passively, whereas a curious person tends to be a bit more active **(curiosity is active feeling**) in their pursuit of acquiring more knowledge." (Participant 84)

"That is, curiosity can be more short term while an interest is usually more long term for the most part. You will often be curious about something, then when you find out more, that initial reaction might fade **(curiosity is active and short term feeling)**. With an interest, it is usually something you are more enamoured with and there is a level there beyond curiosity **(interest is stable and a long term feeling relating to in-depth information)**. It is not something that can be quenched by a simple answer." (Participant 92)

The labelling of codes aimed to capture the differences and similarities of curiosity and interest. In phase 3, we collated codes into potential themes, which reflected major features and patterns in the data. We utilized a mind map to visualize the relationship of the codes, which helped us come up with appropriate themes. To ensure the validity of codes and themes, during this phase the coder (the first author) met other co-authors (fourth and fifth authors) who were not involved in coding, and discussed the appropriateness of the coding process. Here we considered and discussed alternative interpretations until reaching a consensus on the interpretation of patterns in the data. In phase 4 and 5, themes were reviewed again by examining all codes and themes collectively, and we also produced a thematic map. Co-authors reviewed the tentative themes [57]. During the review process, we considered and discussed alternative interpretations until reaching a consensus on the interpretation of patterns in the data. In the last step, phase 6, we determined final themes and identified quotations illustrative of each theme.

## Results

### Overview of themes

The participants' interpretations were captured in two main themes regarding the description of curiosity and interest: (1) curiosity is an active feeling and interest is a stable feeling; (2) curiosity is directed toward uncertainty and interest is directed toward certainty when you want to learn (see Fig 1). Each theme highlighted a major aspect of the respondents' perspectives on the differences of curiosity and interest; however, there were areas of conceptual overlap, which we will discuss later. Below, we provide an explanation of the themes, along with example quotes.

**Fig 1. Diagram showing the transcript topic (level 1), themes (level 2), sub-themes (level 3), and higher-level codes (level 4) [49].**

## 1. Theme: Curiosity is an active feeling and interest is a stable feeling

One theme that arose from participants' responses is that curiosity is an active feeling whereas interest is a stable feeling. The data also indicated that curiosity and interest include common emotions during learning.

> "Curiosity seems like more of an active concept that is impelling you to do something while interest is more of a passive condition." (Participant 109)

> "People who are curious try and go and learn about the world, and actively try to engage with other people to help satisfy their thirst for knowledge." (Participant 63)

> "Interests can also be passive, where someone is content to allow the information to come to them passively, whereas a curious person tends to be a bit more active in their pursuit of acquiring more knowledge." (Participant 2)

> "Curiosity: You take a risk to expand your knowledge. Basically, it's an action that you actively do in order to expand your current knowledge from what it was before." (Participant 77)

> "Curiosity is what you do when you are doing something active about an interest. Also, curiosity seems to be a stronger feeling than interest. If you are curious about something you are more likely to actually do something, while just being interested seems like it would be easier to ignore at times." (Participant 118)

**Sub-theme: Curiosity is the first feeling when you want to know something.** Many participants indicated that curiosity was a general active feeling when people acquire information/knowledge and the active feeling generally was associated with eagerness, drive, pursuit, and desire, etc. In addition, participants considered that curiosity represents an initial feeling with active motivation to obtain information or knowledge.

> "Curiosity is typically not a passive feeling, but rather an active pursuit of knowledge or information, so a critical component is that the individual who feels curious must feel compelled or driven." (Participant 83)

> "Curiosity appears to be an imaginative and eager emotion which leads one to greater knowledge and broader horizons." (Participant 34)

> "You could say that curious comes first then interest. Both the interest and curiosity is about discovery." (Participant 3)

> "I think curiosity is what first gets someone involved in a subject and your interest is what keeps a person involved in that subject over a long period of time." (Participant 90)

Furthermore, participants described the mental mind-set of curious people and argued that curious people think differently and actively without being controlled by external forces.

> "I am very curious in education. It makes my mind thinking differently. It's no secret that curiosity makes learning more effective and enjoyable. Curious students not only ask questions, but also actively seek out the answers." (Participant 50)

> "It often is good to be curious because it keeps your mind active and you will constantly be thinking about new things and different things and how they all interact. "(Participant 52)

> "That uncontrollable sensation that you want to go figure something out, often ignoring the consequences of doing so." (Participant 78)

Participants also indicated that curiosity was different for adults and children, and that curiosity has a child-like nature. Additionally, they described curiosity as a fleeting feeling for information/knowledge. There seems to be a clear connection between the child-like nature of curiosity and other definitions of curiosity such as active thinking and being a fleeting feeling —these characteristics are generally observed in children's behavior when exploring environments (i.e. children's attention changes rapidly in response to external stimuli and they tend to get bored quickly).

> "I feel that many more children express curiosity, as opposed to adults. There is also much more of an openness in children which I feel leads to such curiosity. Whereas, with adults we tend to assume we already know enough about just about everything." (Participant 34)

> "I feel, is greatly important to both children and adults alike. A healthy interest can give one a sense of purpose." (Participant 34)

> "Curiosity can be fleeting, whereas interest is more often sustained." (Participant 6)

Another point that emerged was that curiosity involves both external and internal feelings. In fact, participants emphasized both the external and internal part of feelings in curiosity.

"We could say that curiosity is something rather external because it was an outside product captured by one of the senses." (Participant 25)

"Curiosity is the inner wondering of how something works, exists or functions." (Participant 112)

**Sub-theme: Sustained feeling of interest in a topic.** On the other hand, interest was considered to be a passive feeling in comparison to curiosity. Note that the "passive" feeling for interest in the responses was not used negatively: it means that interest is a stable feeling that supports continuous and deep exploration of information without distraction through boredom and other external stimulations in the long term (see the quotations in the Data Analysis). In addition, participants suggested that interest is a form of curiosity, but it is deeper than curiosity. As can be seen in these interpretations, "passive" feelings of curiosity described by participants had positive connotations; interest was defined as a form of curiosity that consolidates your attention and curious behavior. The passivity of the feeling of interest also helps people spend a long time attending to a topic of interest. In other words, interest is defined as a more stable and slow process that would eventually lead to the deep understanding of information.

"Interest is a passive action that is innate in a person and unique to them". (Participant 41)

"Curiosity and interest are similar in the fact that they both define learning about something. Curiosity and interest differ in the sense that curiosity is usually caused more from being nosy and feeling a need to learn; whereas interest is the actual "want" to find out more about something or learn about something."(Participant 1)

"I personally believe that curiosity and interest are very similar. But, I believe that curiosity is stronger than interest in most cases. Curiosity makes you want to dive in to a thing for subject and learn more. Where interest may be a much milder version of curiosity and you may choose to never take any action on the thing you are interested in at that time. I believe you are more likely to take action about something if you are curious."(Participant 7)

"Curiosity is learning about something we may become interested in. I'm a curious person and research things I've never heard of or don't understand. That's curiosity. But pursuing things further after learning about them is creating interest in them and being interested in them. Curiosity can be fleeting, but interest generally demands more time and effort. I was curious about how to can tomatoes."(Participant 10)

"Without curiosity we would all just be content in our lives and never develop an interest in anything of the world that we are all a part of. Curiosity is the gentle nudge toward interest. You could thing of interest as the effect and curiosity the cause." (Participant 19)

"Interest is a form of curiosity that manifests in the interested person being alert and paying attention to a topic, idea, or activity." (Participant 6)

Moreover, participants gave an analogy with respect to time. From their collective viewpoint, curiosity is a short-term feeling whereas interest is a long-term feeling. This observation is consistent with the other description (mentioned above) that curiosity is a fleeting feeling.

"That is, curiosity can be more short term while an interest is usually more long term for the most part. You will often be curious about something, then when you find out more, that initial reaction might fade. With an interest, it is usually something you are more

enamoured with and there is a level there beyond curiosity. It is not something that can be quenched by a simple answer." (Participant 92)

**Sub-theme: Motivation and positivity as common features in curiosity and interest.**
Participants indicated that curiosity and interest were both innate feelings and positive emotions (i.e., these were common features). Positive emotions were described in terms of the consequences of knowledge acquisition, experiences and actions.

"Curiosity is the natural compulsive behaviour of wanting to know about the world. It helps us to learn and to grow when we are younger and figure things out for our own without parents, siblings, or others teaching us things themselves. "(Participant 12)

"Interests are something that are not learned, but natural. This means that just because someone else is interested in a certain thing or topic." (Participant 41)

"They're both usually connected to positive emotions and things that drive people along a certain path of action. They might be different in that curiosity is more of a compulsion that I don't think can be controlled, whereas an interest is more of a general, vague sense that is somewhat optional and doesn't push you to act with the same level of intensity." (Participant 107)

"Being interested in something is typically a positive feeling in which the person will feel good or benefit from gaining additional information about something. Also, interest tends to imply that a person already has at least some small degree of knowledge about the topic they are interested in." (Participant 5)

Participants also noted that curiosity and interest are both motivators for knowledge acquisition.

"Interest is the state of being motivated to learn about something, to satisfy some personal need or desire." (Participant 13)

"Curiosity has caught your attention and fancy. You think about it and you are motivated to take action." (Participant 21)

## 2. Theme: Uncertainty for curiosity and certainty for interest when you want to learn

This theme captures the different types of approach (i.e. uncertainty orientation and certainty orientation) during knowledge acquisition or information seeking between curiosity and interest.

"To be curious about something, you don't have a clue about it but if something interests you, you have a reason to be interested because some type of thing caught your attention." (Participant 16)

**Sub-theme: You are curious when you want to try new things.**   Curiosity was commonly associated with risk-taking behavior and motivation for trying new experiences. Relatedly, participants indicated that curiosity is a personality trait that is related to open-mindedness, gaining knowledge, and personal development.

"Most people are curious about new information they have learned or curious about something new they have seen." (Participant 19)

"Being curious means having an open mind and seeking more information about something" (Participant 83)

"Curiosity involves risk taking. Interest, on the other hand, is about how creatures are drawn to situations and objects because they have experienced something like them before." (Participant 23)

**Sub-theme: Interest as a strong personal preference (e.g., hobbies).** Although interest was sometimes interpreted as a form of curiosity which serves as an initial motivator of information-seeking, participants further noted that interest focused more on task engagement itself. Participants considered interest to be something that requires action to reengage with a topic, and represented more than simply acquiring a piece of information. Interest is referred to as involving sustained actions to know or learn deeply. They associated interest with something akin to hobbies, relationships and things they liked; all supporting the idea that interest involves long-term engagement and sustained actions. In contrast to curiosity, participants considered that there was little risk-taking associated with interest.

"Curiosity can be quickly forgotten but when people are interested, they are much more active in pursuing the subject." (Participant 61)

"Interest is when you find yourself constantly thinking about something and you want to know more. Your mind easily turns to that subject, and hearing or reading about it is exciting or at least it takes up a lot of you." (Participant 53)

"Interest is when you want to do more than learn about something; it is when you want to engage with the subject of your interest, as opposed to just learning about it or answering a question regarding it." (Participant 46)

"Interest may also relate to hobbies, relationships, likings, etc. Interest may also lead someone to be more curious into something and how they really want to think about it compared to others." (Participant 88)

**Sub-theme: Curiosity involves wanting basic information, interest involves wanting deep knowledge.** Participants typically associated curiosity with a feeling elicited by unknown information. Also, curiosity was related to a search for some missing knowledge or the solution to a mystery.

When participants compared curiosity and interest in terms of its relations to learning, interest was associated with a deep understanding and in-depth thoughts about information; conversely, curiosity was more related to the pursuit of a simple answer and immediate knowledge.

"..but curious is a stronger desire to know about something that it may be just a simple answer. Interest may be something deeper and wanting to know more and more about things."(Participant 108)

**Sub-theme: Knowledge gain as a common aspect of curiosity and interest.** A common theme of interest and curiosity was the process of gaining knowledge through seeking information. In addition, participants indicated that both terms were understood as essential for learning.

"Curiosity is the desire to gain knowledge and information about any given topic."(Participant 6)

"Interest is the act of seeking out the answer of something, not being satisfied with what you see on the surface and wanting to know more of what lies within it." (Participant 46)

"Curiosity and interest are similar in the fact that they both define learning about something." (Participant 19)

## Connection between the themes

We considered potential connections between the two themes. Theme 1 encapsulated the different feelings (active/stable feeling) of curiosity and interest when acquiring knowledge. Theme 2 mainly focused on the feelings relating to anticipating/receiving information (certainty/uncertainty). This suggests that theme 1 and theme 2 seem to capture different stages of the same information-seeking process.

## Discussion

This study aimed to understand naïve participants' beliefs about curiosity and interest. Although the terms are often considered to be separate concepts, the subjective experience of each term by naïve participants has not been examined in the literature—while experts and self-report scales defined these terms based on their own theoretical perspectives, relatively little is known about their natural meaning, discourse, and how they are described by naïve participants. The present results indicate that people ascribe somewhat distinct experiences to curiosity and interest. For example, while curiosity is considered to be active feelings towards uncertainty, interest is considered to be a more stable feeling than curiosity, which is more oriented towards certain things. We also found substantial overlap between the terms, e.g., both terms are closely related to knowledge acquisition process. In addition, the generated themes in the present research may help researchers establish agreed-upon scientific definitions that do not considerably deviate from people's naïve understanding. This is not a trivial issue for applied researchers who are in constant communication with the general public.

Our results also suggest that curiosity is an active feeling which is further characterized as active thinking, a fleeting feeling that is the first feeling one experiences when confronted with an information gap, and a child-like emotion. This characterisation of curiosity supports previous theories of curiosity. Regarding curiosity as an active feeling, Berlyne classified curiosity with a four-way categorization in two dimensions (epistemic-perceptual curiosity and diversive-specific curiosity) [24]. These dimensions implicitly include active feelings, e.g., desire for change, seeking of stimulation, boredom, novelty and desire for knowledge. While Berlyne did not consider the intensity and frequency of feeling, the State-Trait Curiosity Inventory (STCI) did (though the item was "I feel mentally active"; [31]). Other researchers focus on similar active feelings of curiosity, e.g., the researchers [58], who proposed that practically curiosity involves acting and thinking differently to provide an intense desire to discover and engage in novel and challenging experiences. Regarding curiosity as a child-like emotion, Jirout and Klahr emphasized that curiosity in children is characterised as a natural feeling to discover the world [59]. Additionally, they state while children's curiosity is instinctive (i.e., not under intentional control), this is different to adults' curiosity, which can be intentionally directed in order to adapt to new situations [59]. Likewise, some researchers have linked curiosity with patterns of infant behavior where they attend to objects with specific physical properties like bright colors, sounds, human face and movement [60] and also novel objects (even when they have little no prior interaction with these objects, [10]).

Emotion theorists claim that while interest serves long-term developmental goals, curiosity relates to novelty and the possibility of actively broadening experiences [61]. The data seems to

be consistent with this perspective. While the data from the present study indicated that curiosity was an active emotion, interest was defined as a passive, sustained and stable feeling. It is worth noting that some participants considered interest as a "passive" feeling as opposed to the active, transient feelings of curiosity. As we code and compare participants responses, we have come to the conclusion that we can interpret this aspect more as referring to the stability of interest. This is consistent with the previous research demonstrating that interest has a motivational function of maintaining engagement with the environment which allows us to adapt to new experiences that we experience through life [61, 62]. However, our interpretation of the data is open to further discussion.

Thematic analysis also revealed that interest was often considered a form of curiosity and was stronger than curiosity. It was stronger in terms of the intensity of desire and engaging for learning or discovering something. In line with previous research, although these terms were theoretically, empirically, and practically different, they were also highly related [8, 62, 63]. Curiosity and interest both fostered undivided attention and engagement towards new information, which complements findings that interest is thought to narrow attentional scope [64]. Our findings suggested that interest could be defined as involving more intense and sustainable attentional focus than curiosity—this idea is consistent with the perspective that interest is an long-term engagement with specific materials during which people enhance their awareness of the self [65].

Interest was associated with certainty about information that someone wanted to learn (i.e., people are interested in information related to something which they know a piece of information about), whereas curiosity was associated with uncertainty (i.e., people are curious about information related to something they do not know about). In the knowledge-gap model [7], curiosity resulted from the realisation that a piece of information was unknown. According to this account, curiosity is discussed in relation to exploration, uncertainty, tolerance of ambiguity, frustration and sensation seeking [66]. Supporting links between uncertainty and curiosity, there is agreement that curiosity is involved in risk-taking, trying new experiences and immersing oneself in situations with potential for new information or knowledge [25, 67, 68] while interest is involved in behavior that interacts with one's current environment [61]. Moreover, interest supports people engaging with a diverse set of experiences long term, allowing people to reflect themselves in their current environment [9]. In contrast, curiosity (as openness to experience, novelty seeking and intrinsic motivation) allows people to grow through exploration [56–66, 69].

Our thematic analysis revealed that both curiosity and interest were understood as essential for learning or acquiring new information. This is in line with the literature on both curiosity and interest. Curiosity is defined as serving to motivate exploration and interaction with new information [56]. Likewise, interest is an emotion directed towards knowledge that motivates learning and exploration [9]. Prior reviews highlight the role of interest in encouraging people to think deeply and use good meta-cognitive skills [6, 9, 70]. Most studies concerning curiosity discuss curiosity in relation to overarching themes of information-seeking and gaining knowledge, e.g., epistemic curiosity [24], interest-deprivation type curiosity [27], and state-trait curiosity [71, 72]. Our results therefore also corroborate existing theories of curiosity, such as those that refer to it as the "cognitive appetite" [73], a "thirst for knowledge" [70] and/or an "appetite for knowledge" [74]. Our research is in line with previous studies that investigate curiosity and interest in terms of willingness to spend cognitive resources learning new information [75], as the main human motivation for learning [9] and academic performance across different learning environments [76].

Although speculative, these overall results suggest that the human knowledge-acquisition process may be organized in two different parts/stages—short-term/temporary information

seeking based on information uncertainty and stable long-term motivation to engage in a task. A comprehensive account of information-seeking behavior requires consideration of both aspects [14]. This is consistent with the discussion based on the reward-learning framework of knowledge acquisition: While the framework supposes that information-seeking behavior is described by a reward-learning process driven by uncertainty, it also indicates that the fact that information is integrated into a person's existing knowledge adds another layer to the process. Specifically, expanded knowledge increases the expected value of new information, sustaining long-term engagement in knowledge acquisition activities (e.g., information seeking). Our results seem to suggest that the former (short-term information seeking) is close to what people call curiosity and the latter (stable long-term information seeking) is close to what people call interest (though the distinction is not that clear in some respects).

## Limitations

This study should be interpreted with several limitations in mind. First, the current study asked participants for definitions of curiosity and interest, not subjective experiences in their daily life. We decided to ask for definitions in order to understand how people define curiosity and interest beyond their subjective experiences. However, it is possible that participants had never actually thought of such definitive differences, and that the responses they provided simply reflected their post-hoc (not in-the-moment) explanations about the concepts [77]. Relatedly, as we asked participants to provide definitions of curiosity and interest simultaneously (and in a fixed order), it is possible that they implicitly tried to differentiate the concepts. While we believe that such a post-hoc explanation provides valuable information about how people understand curiosity and interest in their daily life, future research should examine the validity of our findings with a different design (e.g., asking independent participants to respond to each of the questions) and/or a different methodology (e.g., ecological momentary assessment [78]). Second, while our study collected data from a relatively broad range of age groups with different ethnic and educational backgrounds, the study is not designed to provide results that can be generalised to broader populations. For example, the sample did not include children or schoolchildren under 18 years old. Post and Walma van der Molen, [45] (discussed in the introduction), found that Dutch children's concept of curiosity was predominantly described in a social domain (e.g., relating to eavesdropping). In hindsight, we found this pattern in a few of the responses in our qualitative analysis, but this was not explicit enough to form a theme. Therefore, examining how children define interest as well as curiosity may be especially revealing. There was also no diversity in nationality with which to evaluate cultural bias because the study was conducted online with U.S. citizens only. Beyond English language, many other languages have distinct words that represent curiosity and interest (e.g., "Neugierde" and "Interesse" in German, "Merak" and "Ilgi" in Turkish, and "Ko-ki-shin" and "Kyo-mi" in Japanese), indicating that the distinction is relatively universal phenomenon. Future studies should examine the generalizability of our findings and potential cultural differences using cross-linguistical comparison.

Third and finally, while online experiments allow us to collect large numbers of responses, it has certain limitations. For example, it is difficult to obtain more in-depth information from participants—future studies may benefit from semi-structured interviews, which allow follow-up questions and the chance for participants to clarify some phrasing, providing an even richer dataset. Semi-structured interviews may also be more appropriate for children, allowing researchers to focus on the child-like forms of curiosity and interest, and potential differences from adult-forms. Furthermore, the online format may prevent participants from in-depth investment in their responses, potentially prompting more shallow-level answers than we were

hoping for. Recent studies have reported sufficient quality for qualitative data collected online [79, 80], and there is an increasing number of qualitative studies using online platforms [81–85]. In addition, participants responses in the current study all seemed sensible to the coder. However, the suitability of online data collection for qualitative studies should be further evaluated in future research.

## Conclusion

This study investigated the naïve beliefs of people about the definition of curiosity and interest. Participants' viewpoints revealed associations between the terms with active/stable feelings and uncertainty/certainty when you acquire new knowledge. Curiosity was considered active, was often equated to a child-like emotion, was a first feeling and a fleeting feeling. Interest was considered a sustainable and grounded feeling. Furthermore, we showed connections between uncertainty and curiosity when acquiring knowledge and found that curiosity was somewhat related to risk-taking behavior, trying new experiences, and wanting basic knowledge in something. In contrast, we showed connections between certainty and interest, and found that interest was related to long-term engagement with something, showing effort and action towards acquiring and sustaining knowledge over time. Although there are differences between curiosity and interest, the terms overlap in that they are both positive emotions, motivations, relate to acquiring knowledge and are both considered essential for learning (i.e. curiosity sparked further interest). Furthermore, curiosity and interest are different for adults and children. Specifically, curiosity and interest encourage individuals to improve and adapt to their environment. It is also clear that the terms are complementary and are likely to work in tandem; curiosity allows individuals to discover information and interest allows the consolidation and maintenance of that knowledge. Participants' interpretations of the terms in their daily life can decrease practically unnecessary confusion regarding the definition of curiosity and interest in applied science (e.g. education, psychology, and neuroscience). This will help researchers to understand and consider peoples' naïve belief about curiosity and interest when designing experiments in this field.

## Supporting information

**S1 Appendix. The phases of thematic analysis.**
(DOCX)

**S2 Appendix. The developed thematic map showing three main themes before the final themes.**
(TIF)

## Author Contributions

**Methodology:** Daniel J. W. Jones.

**Project administration:** Kou Murayama.

**Software:** Greta Fastrich.

**Supervision:** Kou Murayama.

**Writing – original draft:** Sumeyye Aslan.

**Writing – review & editing:** Ed Donnellan, Daniel J. W. Jones, Kou Murayama.

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
