## [Decision Letter · Decision Letter 0]

27 Nov 2020

PONE-D-20-25903

People’s naïve belief about curiosity and interest: a qualitative study

PLOS ONE

Dear Dr. Aslan,

Thank you for submitting your manuscript to PLOS ONE. After careful consideration, we feel that it has merit but does not fully meet PLOS ONE’s publication criteria as it currently stands. Therefore, we invite you to submit a revised version of the manuscript that carefully and systematically addresses all the points raised by the two reviewers of your manuscript during the review process. Both reviewers provide detailed points that you need to address satisfactorily. Please pay particular attention to establishing a more convincing rationale for this research (especially in relation to pertinent previous research), and ensuring that the research aims that you indicate and the methods you use correspond to each other (Reviewer 2 has noted a number of serious problems with the method you used). You also need to ensure that explanations and descriptions of the qualitative analysis procedures you used are sufficiently understandable even to those who do not possess expertise in those procedures.

We look forward to receiving your revised manuscript.

Kind regards,

Emmanuel Manalo, PhD

Academic Editor

PLOS ONE

Journal Requirements:

Reviewers' comments:

Reviewer's Responses to Questions

**Comments to the Author**

1. Is the manuscript technically sound, and do the data support the conclusions?

Reviewer #1: Partly

Reviewer #2: No

2. Has the statistical analysis been performed appropriately and rigorously? 

Reviewer #1: I Don't Know

Reviewer #2: N/A

3. Have the authors made all data underlying the findings in their manuscript fully available?

Reviewer #1: Yes

Reviewer #2: Yes

4. Is the manuscript presented in an intelligible fashion and written in standard English?

Reviewer #1: Yes

Reviewer #2: Yes

5. Review Comments to the Author

Reviewer #1: I enjoyed reading about this study and found it to present interesting data that researchers could use in considering the constructs of curiosity and creativity in future work. One challenge in thinking about the paper’s usefulness, however, was the framing of the work and the discussion of specific ways that it could inform future work. Specifically – why is it useful to know what laypeople think about curiosity and interest, and in what ways can researchers use the themes identified to inform their work? Or – perhaps the paper could be reframed to more directly present the goals of simply understanding how people view curiosity and interest? This latter suggestion seems like one that makes sense to me but is not aligned with the current framing of the literature reviewed and discussion of the results. There are many papers on teachers’ perceptions of student characteristics that do this, and I can think of papers that look at perceptions of curiosity, which is not the same as what this study did but similarly looks at the “layperson perspective”. A couple that come to mind are Kashdan’s paper on how curious people are viewed, and Post’s paper on conceptions of curiosity (citations provided below).

Overall, I think this paper presents interesting data but would be strengthened by reframing the intro around clearer goals of what is learned and how this new knowledge can be used, and then continuing to describe how the knowledge advances specific literature and can be used in different ways in the discussion. I describe a specific suggestion and/or questions related to each section below.

Introduction:

After reviewing prior definitions and theories of curiosity and interest in some depth, with an emphasis on the lack of agreement in definitions and measures, the researchers discuss the reward-learning framework and the purpose of the study: to understand the culturally agreed-upon commonalities and differences between curiosity and interest in layperson definitions. While I believe this purpose is interesting in terms of comparisons to and understanding how scientific research has defined and measured curiosity and interest, it seemed strange that the introduction did very little in terms of discussing prior work that has looked at layperson definitions of these constructs, and why it matters how people outside of the scientific research community define them. The challenge of advancing research without common definitions is given, but what is the value in understanding the layperson definitions? I can imagine many possible answers to this and expect that there is prior research that could be discussed to explain, beyond what is given as justification for the methods used. What is the value of studying naive beliefs?

Methods:

The authors state that they planned to collect 150 participants before examining the data but seemed to have stopped at 135 at which point the data already showed thematic saturation – this was unclear (why were the data examined early?).

I am not a qualitative researcher and appreciated the information about the qualitative methods for identifying themes.

Results:

The diagram was helpful in understanding the structure and organization of responses within the identified themes, and the quoted examples were helpful in demonstrating the themes. I did wonder how common different types of responses were and how variable the responses were. For example, what portion of responses fell into each of the different themes, or maybe it makes more sense to know what portion of responses were outside of those themes? I imagine a single person’s response could touch on multiple themes, but I wondered how many people had responses that were outside of the identified themes, and perhaps how many touched on only one whereas how many were more inclusive of all of the different subthemes? Were there any patterns related to individual differences such as gender, age, or other background characteristics?

Discussion:

I appreciated the contrasting between layperson definitions of curiosity and interest and the ways these have been defined in scientific research in the discussion, but I still felt like the paper could be strengthened with a clearer framing of what can be done with the knowledge learned from this work. The authors state, “Participants’ interpretations of the terms in their daily life can decrease practically unnecessary confusion regarding the definition of curiosity and interest in applied science (e.g. education, psychology, and neuroscience). This will help researchers to understand and consider peoples’ naïve belief about curiosity and interest when designing experiments in this field.” In what ways can the data collected and themes that emerged decrease this confusion? What does that look like – what should future research do as a result of knowing this? How will this help with designing future experiments?

Citations mentioned above:

Kashdan, T. B., Sherman, R. A., Yarbro, J., & Funder, D. C. (2013). How are curious people viewed and how do they behave in social situations? From the perspectives of self, friends, parents, and unacquainted observers. Journal of personality, 81(2), 142-154.

Post, T., & Walma van der Molen, J. H. (2018). Do children express curiosity at school?: exploring children's experiences of curiosity inside and outside the school context. Learning, Culture and Social Interaction, 18, 60-71.

Reviewer #2: The objective of the present research is to examine lay conceptions / naïve beliefs about curiosity and interest. I really like the idea of that research. That said, I do have questions and serious concerns.

Theory

1.) l. 49-50: “lack of consensus about how researchers should conceptualizes these terms”. I don’t agree with that conclusion. Apart from that I think that it is necessary to underline that with literature.

2.) l. 62-64: As a reader I would like to have some literature supporting the statement.

3.) l. 85-86: “curiosity is conceptualized as immediate experience in response to stimuli in the external environment”: I don’t agree with that conclusion. At least epistemic curiosity is elicited also by conceptual problems, puzzling ideas and so on. This has clearly an internal focus. Also, Berlyne distinguished between specific and diverse curiosity behavior. Diverse curiosity behavior is an explorative behavior that is executed because of absent stimulation.

4.) I have serious concerns about your theoretical reasoning. To my opinion, your reasoning for the research question is not sufficiently elaborated: Why would you expect differences in the lay conceptions between curiosity and interest? May be people just use it interchangeably? What exactly is the additional value in knowing the naïve beliefs of lay people? Lay conceptions also have their pitfalls (not explicit, consistent, parsimonious and productive enough).

5.) l. 136: Why there is an inherent impossibility to define these terms in a scientific manner?

Method

6.) There is discussion about the quality of data gained through MTurk (Chmielewski & Kucker, 2019; Buhrmester, Talaifar & Gosling, 2018). I think you should at least discuss possible impact on data quality; also consider how that discussion applies to qualitative data.

7.) I wonder whether you have also captured educational background and English literacy. Answers on open question certainly depend on the eloquence of participants.

8.) l. 169: you mentioned pilot data? What kind of pilot study did you conduct?

9.) I have serious concern about the fit between your research aim and the method you apply, or more specifically the question you ask the participants:

- In line 135 – 136 you mention that “people have intuitive, but somehow agreed-upon understanding of how they are different”. However, if you target the intuitive concepts of people it is not appropriate to as them for an explicit definition. In my opinion you should than analyze how people are using these two concepts in their everyday language.

- People don’t have a ready definition of interest and curiosity in their mind.

- In the discussion you emphasize that “the use of open-ended questions ensured that no conceptual priming took place” (l. 483). I strongly disagree with that. By posing one question for curiosity and one for interest you imply that these are two different things. And you intensify that by asking about similarities and differences.

10.) As a reader I like to know more about the answers given: How much did the participants write? did everybody answered to every question? Did somebody write that the concepts are the same?

11.) I think more information about the coding process are necessary. How did you come to your themes? As a reader I would like you to illustrate the different steps with examples. Until now the reproducibility is low.

12.) Are there any parameters that provide information about the quality of the coding process?

13.) l. 205 what does “line by line coding” mean? Is it a sentence?

14.) What do you mean by “we coded the explicit and implicit meaning for all data”? please give an example.

Results:

In general, I have difficulties understanding the results. The meaning or content of the themes and subthemes are not sufficiently clear and precisely described.

15.) If I understood right: in the first theme curiosity and interest is ordered along two dimensions, namely, passive – active and stable – fleeting. Is that correct?

16.) In l. 280 – 283 you equate passive with stable. I don’t see in your quotation why that should be like that? (l. 231-232)

17.) I sometimes don’t understand how the quotations fit to theme or sub-theme (e.g. sub theme l. 311 -320: these quotations don’t fit to the part positive emotion?)

18.) L. 378: “curiosity is more related to the pursuit of basic and immediate knowledge”; what do you mean by basic? In the sense of fundamental? Or in the sense of simple?

19.) L. 393 – 395: I don’t understand that paragraph.

20.) Your figure is difficult to read (it is pixelated).

Discussion

I find it a bit difficult to evaluate the discussion, as I am not really clear with the results.

21.) In my opinion the authors should broaden the limitation part and discuss about the

- possible data quality constraints due to mTurk

- implicit theories, lay theories are known for contrariness, inconsistencies, lack of explicitness and so forth. You should discuss about that in relation to your data (e.g l. 290 “passive action”; what is meant by child-like emotion?)

22.) How does the knowledge gained through the present study does contribute to the differentiation between interest and curiosity from the scientific perspective.

23.) l. 440-442: as far as I know the literature, for adults and for children curiosity does not arise if something is too new, complex etc. and it does not arise when it not new, complex etc. at all.

24.) l. 449 – 450: “more intense attentional focus than curiosity”: in the results interest is described as passive; how does that matches?

6. PLOS authors have the option to publish the peer review history of their article (what does this mean?). If published, this will include your full peer review and any attached files.

Reviewer #1: No

Reviewer #2: No

---

## [Author Response · Author response to Decision Letter 0]

15 Apr 2021

Thanks for your comments. They were helpful to improve the manuscript.

---

## [Decision Letter · Decision Letter 1]

17 Jun 2021

PONE-D-20-25903R1

People’s naïve belief about curiosity and interest: a qualitative study

PLOS ONE

Dear Dr. Aslan,

Thank you for submitting your manuscript to PLOS ONE. After careful consideration, we feel that it has merit but does not fully meet PLOS ONE’s publication criteria as it currently stands. Therefore, we invite you to submit a revised version of the manuscript that addresses the points raised during the review process. In particular, we would like you to carefully and systematically address ALL the outstanding issues detailed by Reviewer 2 below.

We look forward to receiving your revised manuscript.

Kind regards,

Emmanuel Manalo, PhD

Academic Editor

PLOS ONE

Reviewers' comments:

Reviewer's Responses to Questions

**Comments to the Author**

1. If the authors have adequately addressed your comments raised in a previous round of review and you feel that this manuscript is now acceptable for publication, you may indicate that here to bypass the “Comments to the Author” section, enter your conflict of interest statement in the “Confidential to Editor” section, and submit your "Accept" recommendation.

Reviewer #1: All comments have been addressed

Reviewer #2: All comments have been addressed

2. Is the manuscript technically sound, and do the data support the conclusions?

Reviewer #1: Yes

Reviewer #2: Partly

3. Has the statistical analysis been performed appropriately and rigorously? 

Reviewer #1: N/A

Reviewer #2: N/A

4. Have the authors made all data underlying the findings in their manuscript fully available?

Reviewer #1: (No Response)

Reviewer #2: (No Response)

5. Is the manuscript presented in an intelligible fashion and written in standard English?

Reviewer #1: Yes

Reviewer #2: Yes

6. Review Comments to the Author

Reviewer #1: Thank you for the responses to my questions and your work revising this manuscript. I think it will be of interest to many others, and I also value having more qualitative research on curiosity.

Reviewer #2: Comments to the authors

Thanks for addressing the issues raised. I think the manuscript has become better. That said, I still have questions and concerns.

Theory

I still have concerns about your theoretical reasoning.

1.) I think you should explain the reward-learning framework of knowledge acquisition. Otherwise the respective paragraph is hard to understand.

2.) I don’t understand how the reward-learning framework of knowledge acquisition helps to explain why it is so difficult to define curiosity and/or interest? I mean a lot of our psychological constructs have been “out there” before psychology started to study them; that, however, does not mean that we don’t have agreed upon definition.

3.) p. 6: I understand your reasoning that if people construed two different terms that it seems likely that there are also two different underlying concepts (even though one could surely also argue that the more important a concept is the more words are construed for that specific concept). Anyway, I would not go with you that they are qualitatively different and are supported by qualitatively different psychological processes. You could call a person bright, brainy, clever, and/or intelligent; and these words certainly have slightly different connotation, however, they are not something totally different, and are probably not supported by qualitatively different psychological processes.

4.) P. 6: Even though I understand you’re reasoning that the somehow “fresh” view of lay people may bring some new insights in the discussion, I wonder why you expect that the scientific definitions of interest and curiosity substantially deviate from lay perception of these concepts?

5.) We know from psychological research that lay beliefs and implicit theories have real-life consequences (e.g. implicit theories about intelligence, illness perceptions). Could you imagine something in the same line?

6.) p.7: What are the results of the studies examining people’s perception about curiosity/interest? Is there anything to infer for the results of the present study? Are the results somehow similar to the results reported here?

Method

7.) I still have concerns about your methods. I would not expect everybody to say that curiosity and interest are the same concepts. However, I wonder if you would have asked different groups of people about curiosity and interest whether the differences would be so pronounced. And I would also expect that you find more commonalities.

8.) Did you randomize the first question?

9.) As far as I could see also in the independent data you cite, people were asked for a definition of curiosity and interest. So actually we have the same situation.

10.) Is there anything like Kappa, interrater reliability?

Discussion

11.) P. 22 You mentioned that the results suggest that the human knowledge-acquisition process may be organized in two different part/stages. What parts of the model/framework do you mean? Or don’t you mean the framework? Could you elaborate more on that.

12.) If you do not mean the framework – could you please discuss your results with respect to the model? To what parts of the model do the lay concepts relate?

7. PLOS authors have the option to publish the peer review history of their article (what does this mean?). If published, this will include your full peer review and any attached files.

Reviewer #1: No

Reviewer #2: No

---

## [Decision Letter · Decision Letter 2]

12 Aug 2021

People’s naïve belief about curiosity and interest: a qualitative study

PONE-D-20-25903R2

Dear Dr. Aslan,

We’re pleased to inform you that your manuscript has been judged scientifically suitable for publication and will be formally accepted for publication once it meets all outstanding technical requirements.

Kind regards,

Zhidan Wang, Ph.D

Academic Editor

PLOS ONE

Additional Editor Comments (optional):

Reviewers' comments:

Reviewer's Responses to Questions

**Comments to the Author**

1. If the authors have adequately addressed your comments raised in a previous round of review and you feel that this manuscript is now acceptable for publication, you may indicate that here to bypass the “Comments to the Author” section, enter your conflict of interest statement in the “Confidential to Editor” section, and submit your "Accept" recommendation.

Reviewer #1: All comments have been addressed

2. Is the manuscript technically sound, and do the data support the conclusions?

Reviewer #1: Yes

3. Has the statistical analysis been performed appropriately and rigorously? 

Reviewer #1: N/A

4. Have the authors made all data underlying the findings in their manuscript fully available?

Reviewer #1: Yes

5. Is the manuscript presented in an intelligible fashion and written in standard English?

Reviewer #1: Yes

6. Review Comments to the Author

Reviewer #1: I found the revision to the section discussing the reward learning framework to be a nice addition to the paper. I believe the authors adequately addressed limitations of this work in the revision, and that the contribution this research makes is clear in the discussion.

7. PLOS authors have the option to publish the peer review history of their article (what does this mean?). If published, this will include your full peer review and any attached files.

Reviewer #1: No

---

## [Editor Report · Acceptance letter]

23 Sep 2021

PONE-D-20-25903R2 

People’s naïve belief about curiosity and interest: a qualitative study 

Dear Dr. Aslan:

I'm pleased to inform you that your manuscript has been deemed suitable for publication in PLOS ONE. Congratulations! Your manuscript is now with our production department. 

Kind regards, 

on behalf of

Dr. Zhidan Wang 

Academic Editor

PLOS ONE